# Reconstitution of the complete pathway of ITS2 processing at the pre-ribosome

Lisa Fromm[1], Sebastian Falk[2], Dirk Flemming[1], Jan Michael Schuller[2], Matthias Thoms[1], Elena Conti[2] & Ed Hurt [1]

Removal of internal transcried spacer 2 (ITS2) from pre-ribosomal RNA is essential to make functional ribosomes. This complicated processing reaction begins with a single endonucleolytic cleavage followed by exonucleolytic trimming at both new cleavage sites to generate mature 5.8S and 25S rRNA. We reconstituted the 7S→5.8S processing branch within ITS2 using purified exosome and its nuclear cofactors. We find that both Rrp44's ribonuclease activities are required for initial RNA shortening followed by hand over to the exonuclease Rrp6. During the in vitro reaction, ITS2-associated factors dissociate and the underlying 'foot' structure of the pre-60S particle is dismantled. 7S pre-rRNA processing is independent of 5S RNP rotation, but 26S→25S trimming is a precondition for subsequent 7S→5.8S processing. To complete the in vitro assay, we reconstituted the entire cycle of ITS2 removal with a total of 18 purified factors, catalysed by the integrated activities of the two participating RNA-processing machines, the Las1 complex and nuclear exosome.

[1] Biochemie-Zentrum der Universität Heidelberg, Im Neuenheimer Feld 328, Heidelberg D-69120, Germany. [2] Max Planck Institute of Biochemistry, Am Klopferspitz 18, D-82152 Martinsried, Germany. Correspondence and requests for materials should be addressed to E.H. (email: ed.hurt@bzh.uni-heidelberg.de)

Eukaryotic ribosomes are formed through a series of consecutive assembly and maturation reactions, which include folding, modification and processing of the pre-rRNA, and incorporation of ~80 ribosomal proteins, which finally leads to mature and functional ribosomes. This complicated process starts in the nucleolus with transcription of a large ribosomal RNA precursor, called 35S in yeast and 47S pre-rRNA in humans, which consists of mature 18S, 5.8S and 25S/28S rRNA, and intervening RNA sequences, 5′ and 3′ external transcribed spacers (5′-ETS, 3′-ETS), and internal transcribed spacers 1 and 2 (ITS1, ITS2) that are removed at distinct points of the pathway[1, 2]. A large number of non-ribosomal factors are involved in these RNA-processing reactions as well as in many other maturation steps, which transiently associate with the nascent pre-ribosomal particles, thereby driving the whole process toward functional 60S and 40S subunits[3, 4].

One key step during large subunit synthesis is the removal of ITS2 from the newly forming pre-60S particles. ITS2 is located between the 5.8S and 25S rRNA moieties, and is eventually removed from a larger rRNA precursor, called 27S pre-rRNA in yeast, by several RNA cleavage and processing factors[1, 3, 5]. ITS2 processing is initiated by a cleavage event at site $C_2$ within ITS2, which creates free 5′ and 3′ ends that are subsequently trimmed by different exonucleases to completely remove the rest of the overhanging pre-rRNA (Supplementary Fig. 1). The final mature ends, which are the 3′ end of the 5.8S and the 5′ end of the 25S rRNA, are not joined, but interact to form the 'proximal stem', a typical rRNA secondary structure of the mature 60S subunit[6].

The Las1 endonuclease has been recently discovered to catalyse $C_2$ cleavage within ITS2, thereby creating the 7S pre-rRNA, which is the precursor of the 5.8S rRNA, and 26S pre-rRNA, which in turn is the precursor to 25S rRNA (Supplementary Fig. 1). Las1 is part of a complex that brings along other factors for subsequent pre-rRNA processing in yeast and human cells[7–9]. One such factor is Grc3, which phosphorylates the 5′ end of the 26S pre-rRNA, enabling the Rat1 exonuclease and its cofactor Rai1 to efficiently trim 26S to 25S′ pre-rRNA. In contrast, 7S pre-rRNA generated by $C_2$ cleavage is further processed by the nuclear exosome, a large RNA-degrading machine exhibiting predominantly 3′→5′ exonuclease activities[10–12].

In almost all eukaryotes, the exosome is assembled from a catalytically inactive ring-shaped nine-component core and Rrp44, which sits on one side of the ring and exhibits both 3′→5′ exonuclease and endonuclease activities[13–15]. The nuclear form of the exosome carries, in addition to Rrp44, another 3′→5′ exonuclease, Rrp6, located on the opposite side of the core[13, 16]. The nuclear exosome has several cofactors, Mpp6, Rrp47 and the ATP-dependent RNA helicase Mtr4, the latter being shown to have a broad role in pre-rRNA processing[17–19]. Whereas Rrp47 and Rrp6 form a composite surface for Mtr4 recruitment[20], Mtr4 is thought to unwind secondary RNA structure elements or remove protein factors from the RNA to make it accessible for the exosome[21, 22]. Moreover, Mtr4 specifically interacts with the ribosome biogenesis factor Nop53, which acts as bridging factor between pre-ribosome and exosome[23]. During 7S→5.8S pre-rRNA processing, Rrp44 trims until it forms a 5.8S + 30 intermediate, which is further shortened by the other exonuclease Rrp6 to an intermediate in the nucleus termed 6S pre-rRNA[24]. 6S pre-rRNA undergoes final processing after nuclear export of the pre-ribosome, catalysed by four non-essential cytoplasmic exonucleases, Ngl2, Rex1, Rex2 and Rex3. This provides a rationale for why ribosomes carrying 6S pre-rRNA are functional[2] (see Supplementary Fig. 1).

Within the architecture of the 60S pre-ribosome, ITS2 is decorated by different ribosome assembly factors, which together form a characteristic structure, initially called the 'foot' that was recently revealed by cryo-electron microscopy (cryo-EM) and shown to contain ITS2[25–27]. Together with data from cross-linking and cDNA analysis, which revealed binding sites of Nop15, Rlp7 and Cic1 on the ITS2 sequence[28, 29], the first

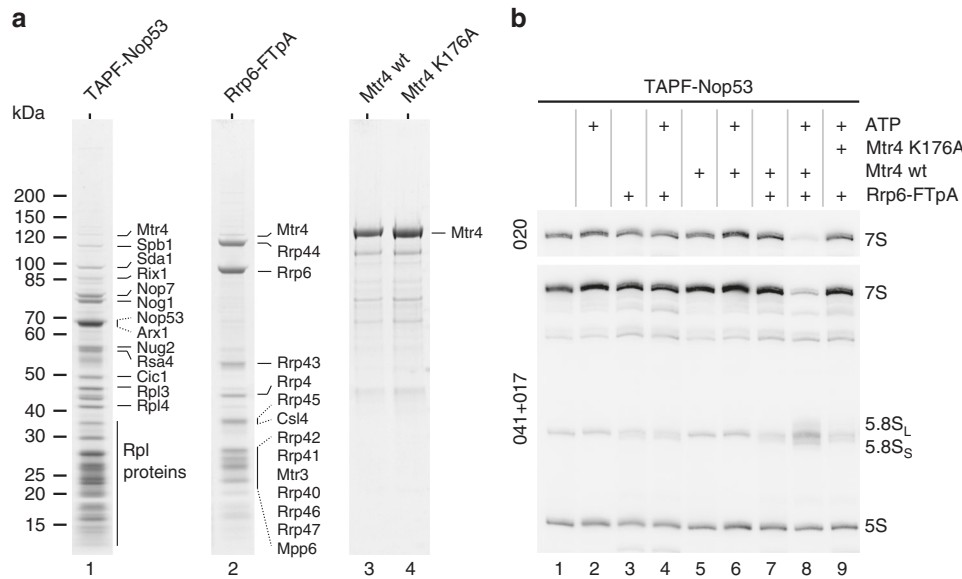

**Fig. 1** In vitro assay for 7S→5.8S pre-rRNA processing in purified pre-60S particles catalysed by the TAP-purified nuclear exosome and cofactors. **a** Analysis of the protein samples used for the in vitro 7S→5.8S pre-rRNA-processing reaction. Substrate 7S pre-rRNA was part of the pre-60S particles affinity-purified via TAPF-Nop53. Endogenous yeast exosome was affinity-purified via Rrp6-FTpA, and recombinant yeast Mtr4, either wt or the K176A mutant, was expressed and affinity-purified from *E. coli*. The indicated eluates were analysed by SDS-PAGE and Coomassie staining. Marked bands of the exosome were identified by mass spectrometry. **b** Northern blot for the detection of 7S pre-rRNA, 5.8S rRNA and 5S rRNA species after the in vitro processing reaction. Processing of 7S pre-rRNA from pre-60S particles (TAPF-Nop53) using the endogenous yeast-derived nuclear exosome (FTpA-Rrp6) and its cofactor Mtr4, either wt or K176A mutant, in the presence or absence of ATP. 7S pre-rRNA was detected with two different probes. 5S rRNA served as a loading control

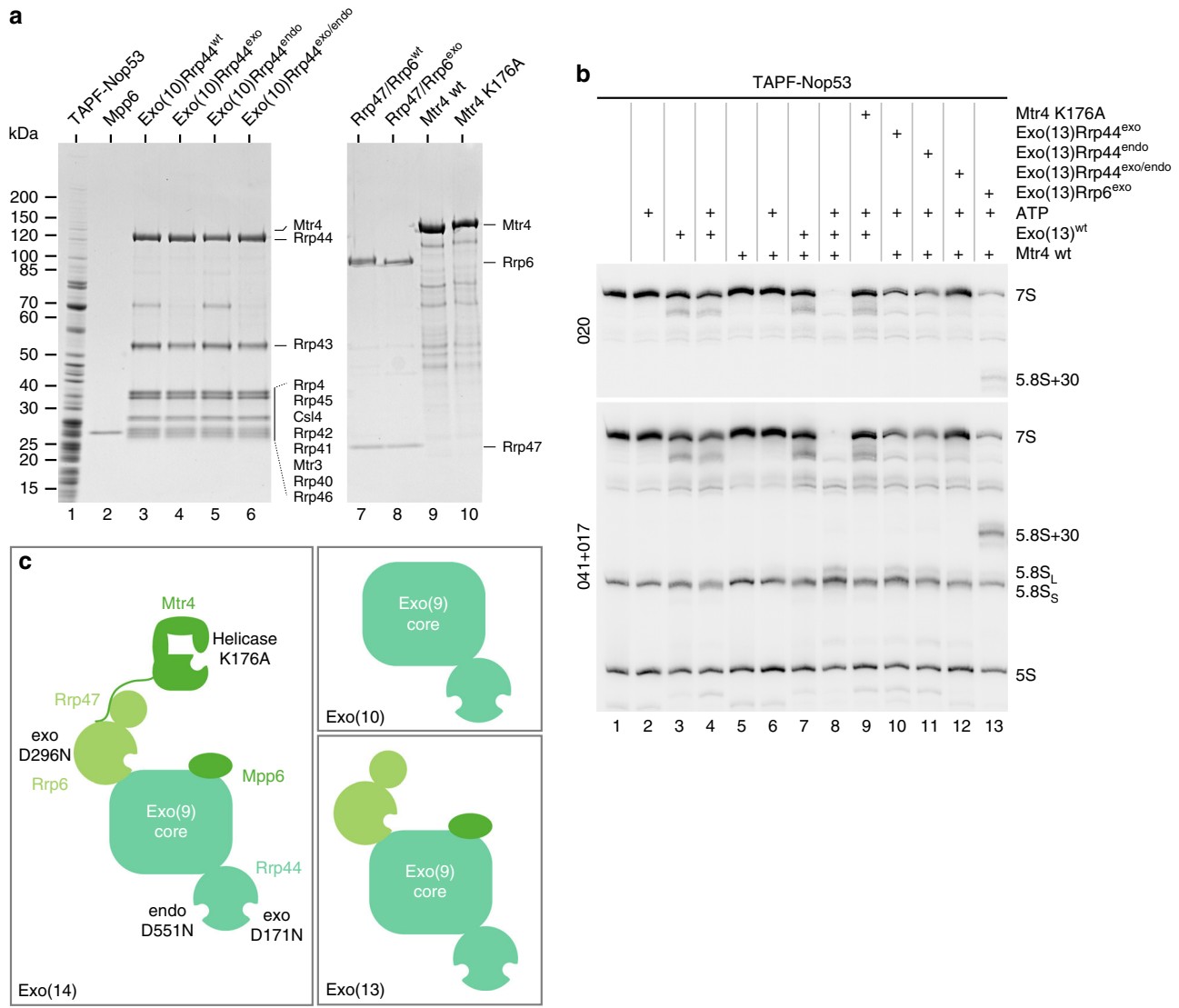

**Fig. 2** In vitro assay for 7S→5.8S pre-rRNA processing in purified pre-60S particles catalysed by the recombinant nuclear exosome and cofactors. **a** SDS-PAGE and Coomassie staining of the protein samples used for in vitro 7S→5.8S pre-rRNA processing. Pre-60S particles containing 7S substrate pre-rRNA were affinity-purified via TAPF-Nop53. All the yeast exosome subunits and Mtr4 were expressed recombinantly in *E. coli* and subsequently purified from cell lysates. Exo(10) comprises the exosome core and includes Rrp44, either in wt or catalytic mutant form, Exo(10)Rrp44$^{exo}$, Exo(10)Rrp44$^{endo}$, Exo(10) Rrp44$^{exo/endo}$, Rrp47 and Rrp6, in mutant and wt form, the latter added as a dimer, and Mpp6 alone. Together, they form Exo(13), which corresponds to the endogenous exosome purified via Rrp6-FTpA. **b** Northern blot for the detection of 7S pre-rRNA, 5.8S rRNA and 5S rRNA species after the in vitro processing reaction. Processing of 7S pre-rRNA from pre-60S particles (TAPF-Nop53) using the recombinantly purified exosome (Exo(13)) in wt and mutant forms and its cofactor Mtr4, either wt or K176A mutant, in the presence or absence of ATP. 7S pre-rRNA was detected with two different probes. 5S rRNA served as a loading control. **c** Schematic representation of the different exosome modules/complexes—Exo(10), Exo(13) and Exo(14)—assembled using recombinant proteins, used in the various in vitro processing assays. Exo(14) also contains the Mtr4 helicase. Further depicted are the enzymatically active subunits of the exosome with their catalytic centres (exonuclease, endonuclease, ATPase) and the mutations generated at these sites

pseudo-atomic models of the nascent 60S subunit isolated via assembly factors Arx1 or Nog2 showed how the ITS2-associated factors Cic1, Nop15, Rlp7, Nop7 and Nop53 constitute the prominent 'foot' structure[26, 27, 30]. Interestingly, another nucleoplasmic pre-60S particle, the Rix1 particle, does not have such a 'foot' structure, suggesting that ITS2 removal can occur between the Arx1/Nog2- and Rix1-affinity-purified pre-60S particle[31].

In this study, we have developed our in vitro assay further to fully excise ITS2 RNA from the pre-60S particle. For this purpose, we affinity-purified pre-ribosomal particles containing the appropriate substrate RNA, either 7S pre-rRNA or 27SB pre-rRNA, and incubated them with purified processing factors.

Analysis of these mixtures revealed that the Las1 complex, in conjunction with the nuclear exosome and its cofactor helicase Mtr4, was necessary and sufficient to catalyse the cycle of ITS2 removal from the nascent 60S subunit. Thus, our analysis gives insight into how two conserved RNA-processing/degrading machineries cooperatively process intermediate RNA from precursor ribosomal RNA in the context of the nuclear pre-60S subunit.

## Results

**In vitro 7S pre-rRNA processing by exosome and cofactors.** In order to understand the mechanisms that underlie the

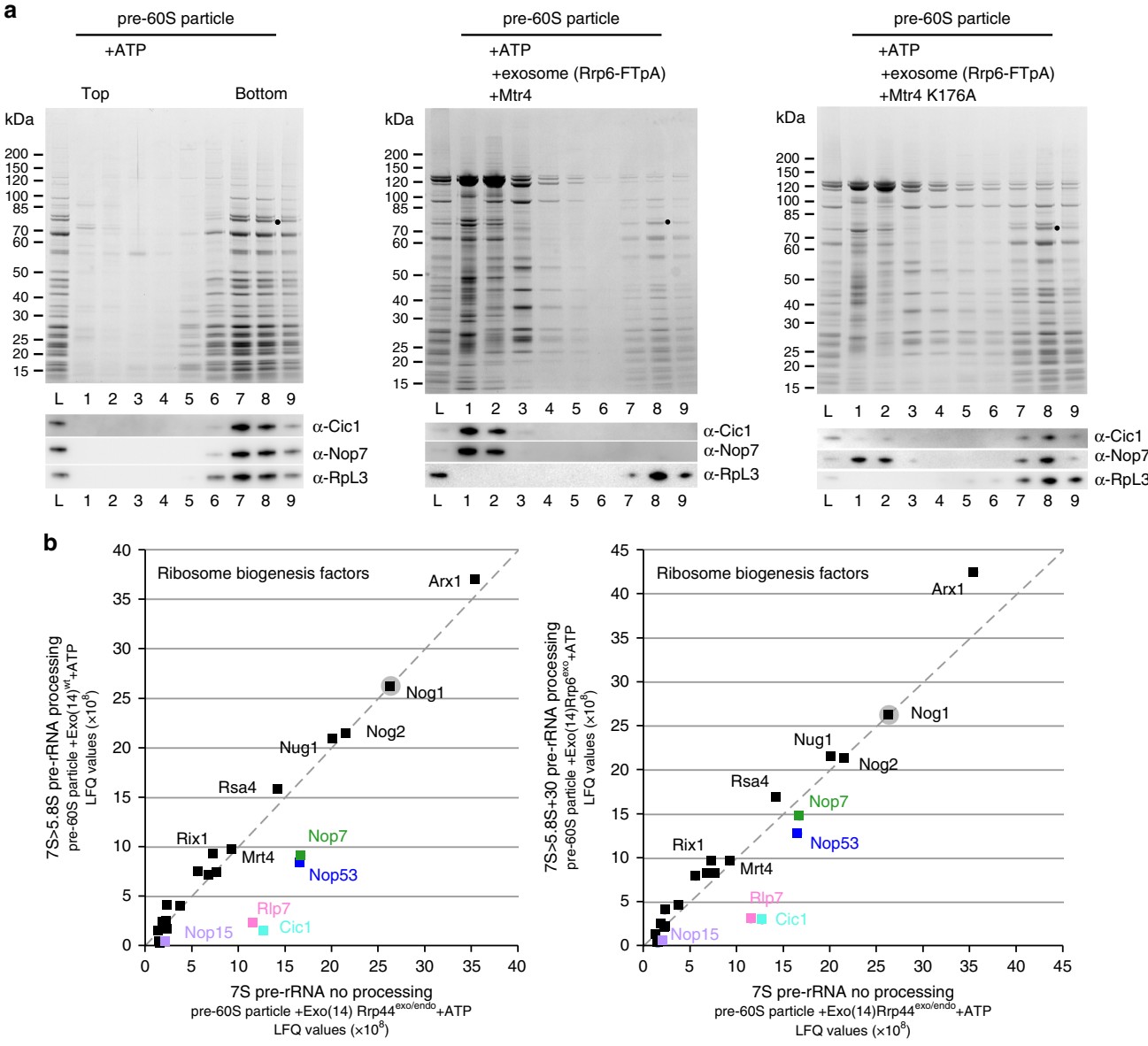

**Fig. 3** Removal of ITS2-associated factors from the pre-60S particles upon in vitro 7S→5.8S pre-rRNA processing. **a** Glycerol gradient centrifugation of pre-60S particles (affinity-purified via TAPF-Nop53) after the in vitro processing reaction using endogenous yeast exosome (affinity-purified via Rrp6-FTpA) and recombinant wt Mtr4 or mutant Mtr4 (K176A) in the presence of ATP. As a control, the pre-60S particle was also incubated only with ATP (mock). After centrifugation, the linear glycerol gradient was fractionated into fractions 1–9, which were TCA-precipitated and analysed by SDS-PAGE and Coomassie staining (upper panel) or western blotting using the indicated antibodies (lower panel), with Rpl3 serving to identify the particle fractions. The bottom fractions (7–9) contained the pre-60S particles, whereas the top fractions (1–2) contained the released factors such as Cic1 and Nop7. Ribosome assembly factor Nog1 not released during the in vitro reaction is indicated by a dot (fraction 8). **b** Semi-quantitative mass spectrometry of pre-60S particle glycerol gradient fraction 8 after 7S→5.8S pre-rRNA processing by recombinant nuclear exosome (see Supplementary Fig. 3b). Label-free quantification (LFQ) values between the different experiments (7S→5.8S pre-rRNA processing, 7S→5.8S + 30 pre-rRNA processing, no 7S pre-rRNA processing) were normalised to Nog1. Afterwards, pairs of LFQ values of ribosome biogenesis factors obtained from different processing stages were compared by plotting them against each other. Left: '7S→5.8S pre-rRNA processing' (pre-60S particle + Exo(14)[wt] + ATP) vs. 'no 7S pre-rRNA processing' (pre-60S particle + Exo(14)Rrp44[exo/endo] + ATP). Right: '7S→5.8S + 30 pre-rRNA processing' (pre-60S particle + Exo(14)Rrp6[exo] + ATP) vs. 'no 7S pre-rRNA processing' (pre-60S particle + Exo(14)Rrp44[exo/endo] + ATP). Labelled in colour are ribosome assembly factors associated with ITS2, which significantly dropped below the dashed reference line ($x = y$), indicating release from the 60S particles. In contrast, most of the other pre-60S factors remained close the reference line, indicating no significant release from the pre-60S particles. All experiments were performed twice with similar outcomes (see Supplementary Table 3)

intertwined processing steps of ITS2 removal from pre-rRNA, we conducted in vitro reconstitution studies. By following this strategy, we were able recently to reconstitute in vitro C₂ cleavage within ITS2 and subsequent 26S pre-rRNA processing, based on purified processing factors, organised within the Las1 complex, and appropriate pre-60S particles containing the cognate

substrate 27SB pre-rRNA[12]. Next, we sought to investigate the processing of the cleaved ITS2 fragment, an ~150-nucleotide long 3′ extension of the 5.8S rRNA, which was not further trimmed in our previous in vitro processing assay. To develop a faithful 7S→5.8S pre-rRNA processing reaction in vitro, we used a pre-ribosomal particle containing predominantly 7S pre-rRNA

purified using the adaptor protein Nop53, known to recruit the Mtr4 helicase and nuclear exosome cofactor to the pre-60S ribosome[23]. Accordingly, we selected the pre-60S particles that co-enrich with Nop53 during tandem affinity purification (Fig. 1a, lane 1).

Our first attempts to develop such a 7S→5.8S-processing assay involved using a nuclear exosome preparation directly isolated from yeast via affinity purification of TAP-tagged Rrp6. This nuclear exosome complex consisted of nine non-catalytic core subunits, the two nucleases Rrp44 and Rrp6, and the cofactors Rrp47 and Mpp6, but only sub-stoichiometric amounts of the Mtr4 helicase (Fig. 1a, lane 2). Hence, we further purified recombinant yeast Mtr4, either the wild-type (wt) or catalytic Walker A motif mutant (K176A), from *Escherichia coli* to supplement the assay with this cofactor (Fig. 1a, lanes 3 and 4). Subsequently, all purified components were mixed in different combinations in vitro with Nop53-purified pre-60S particles as RNA substrate. After incubation, the reaction was analysed by northern blotting and we observed a clear 7S→5.8S pre-rRNA-processing reaction with disappearance of the 7S signal and increase of the 5.8S signal. The generated 5.8S band, however, was somewhat 'smeared', suggesting that the 3′ end of the generated 5.8S rRNA might not be nucleotide-precise (Fig. 1b, lane 8; see also the Discussion). Apparently, the observed 7S→5.8S trimming was dependent on the presence of all three constituents - the nuclear exosome, Mtr4 and ATP (Fig. 1a, b)—and if one of these components was omitted or if a catalytic Walker A motif Mtr4 mutant (K176A) was used, 7S pre-rRNA processing was inhibited, consistent with earlier in vivo data[32]. Thus, we had successfully set up an in vitro assay for 7S→5.8S pre-rRNA processing.

**Concerted action of Rrp6 and Rrp44 in 7S pre-rRNA processing**. To gain insight into the interplay of the different nucleases associated with the nuclear exosome during 7S pre-rRNA processing, we switched to a recombinant exosome system. This has the advantage of avoiding contamination with other yeast proteins and allows testing of exosome complexes with incorporated inactive catalytic subunits. As a first module, we generated a minimal exosome (called Exo10) with all the core subunits plus Rrp44 that was either wt or mutated in its exonuclease (D171N), endonuclease (D551N) or both activities (D171N/D551N). The second module added to the core exosome was the Rrp6–Rrp47 dimer, which enabled us to switch between wt and exonuclease-deficient Rrp6 (D296N)[33–35]. Finally, Mpp6 was added as last subunit, leading to the reconstitution of the entire, that is, the 13-subunit-carrying nuclear exosome, similar to the endogenous exosome (see above and Fig. 2a, b).

Comparable to the endogenous exosome, the recombinant wt exosome (Exo13) efficiently processed 7S→5.8S pre-rRNA in vitro, again based on the Nop53 pre-60S particle that served as an rRNA substrate (compare Fig. 2b with 1b, lanes 1–9). Wild-type Exo13 supplemented with mutant Mtr4 (K176A) mutant did not effect 7S→5.8S pre-rRNA processing, supporting that we had faithfully reconstituted this branch of the ITS2-processing pathway, purely based on recombinant factors (Fig. 2b, lane 9). Notably, only the Rrp44 double mutant efficiently blocked 7S pre-rRNA processing, whereas the single Rrp44 mutants, defective in either exo- or endonuclease activity, allowed only partial 7S→5.8S pre-rRNA processing (Fig. 2b, lanes 10–12; for quantification of bands see Supplementary Fig. 2a). Moreover, if the Rrp6 catalytic mutant was assembled into Exo13, we observed a partial trimming of the 7S pre-rRNA to the 5.8S + 30 intermediate (Fig. 2b, lane 13), which has also been observed in vivo using *rrp6* mutants[24]. These data could be independently reproduced in an assay using a recombinant exosome that contained different

Rrp44 mutations in combination with mutated Rrp6 (Supplementary Fig. 2b, c). Together, these findings demonstrated that initial processing by Rrp44 generates a 5.8S + 30 intermediate, which can only be efficiently blocked by the Rrp44 double mutant (Supplementary Fig. 2a, b). Finally, the in vitro assay also showed faithful hand over of the partially processed 5.8S + 30 pre-rRNA from Rrp44 to Rrp6.

**Release of ITS2-associated factors during 7S processing**. We next investigated the biochemical and structural changes of the pre-60S particle undergoing in vitro 7S→5.8S pre-rRNA processing. For this purpose, we performed glycerol gradient centrifugation of the entire reaction mixture to separate assembly factors that might dissociate from the pre-60S particle during in vitro processing. As anticipated, the Nop53-purified pre-60S particles in the absence of exosome and cofactors (mock control) migrated toward the bottom of the glycerol gradient, where the associated ribosome assembly factors were also detected (Fig. 3a, left panel). Among them are the 'A3 cluster' factors Cic1/Nsa3, Nop15, Nop7 and Rlp7, which, together with Nop53 and 7S pre-rRNA (derived from ITS2 processing), constitute the prominent 'foot' structure of the nuclear pre-60S particles[27, 28] (see Supplementary Fig. 3b). Another factor typically associated with these pre-60S particles is Nog1, the main folded domain of which is, however, located in a different area of the pre-60S particle, close to Rsa4 and Nsa2[30]. Strikingly, if pre-60S particles were incubated with yeast-purified exosome, Mtr4, and ATP to induce 7S→5.8S pre-rRNA processing, the pre-60S particles still migrated towards the bottom of the glycerol gradient. In contrast, Cic1 and Nop7 were dissociated and found on top of the glycerol gradient (Fig. 3a, middle panel). However, if wt Mtr4 was replaced by mutant Mtr4 (K176A) in the otherwise identical reaction mixture, Cic1 and most of Nop7 remained bound to the pre-60S particles (Fig. 3a, right panel).

Similar results were obtained if the in vitro processing reaction was performed with the recombinant exosome (Exo14 + ATP) and subsequently analysed by glycerol gradient centrifugation (Supplementary Fig. 3b). In the case of the fully active Exo14, Cic1 was again efficiently released from the pre-60S particle, but not if the Rrp44 double mutant was used instead (compare Supplementary Fig. 3b, upper left and right panels). Notably, the Rrp6 mutant, known to stop at the 5.8S + 30 intermediate (Supplementary Fig. 1), still induced Cic1 dissociation, but Nop7 remained considerably more associated (Supplementary Fig. 3b, lower left panel). The apparent ITS2-associated factor release observed during the in vitro 7S→5.8S pre-rRNA processing reaction was confirmed by semi-quantitative mass spectrometry, suggesting that, in particular, assembly factors Cic1 and Rlp7, which have direct contact to ITS2 and are exclusively associated with the 'foot' structure, were most efficiently released during the in vitro 7S→5.8S pre-rRNA processing reaction (Fig. 3b, Supplementary Fig. 3a).

**Foot structure removal occurs upon in vitro 7S processing**. To directly visualise the structural changes of the pre-ribosomal subunit undergoing in vitro 7S→5.8S pre-rRNA processing, we subjected these in vitro matured particles, after an ultra-centrifugation step, to negative-stain EM. Nop53 substrate particles incubated only with ATP (mock control) exhibited the typical structural features previously observed for related pre-60S particles with prominent 'foot' (part of ITS2 and associated factors), 5S RNP and Arx1 structures (Fig. 4, compare left and right panels). However, after the in vitro processing reaction catalysed by the yeast-derived exosome and its cofactor Mtr4 in the presence of ATP, most of the particles (~90%) lost their 'foot'

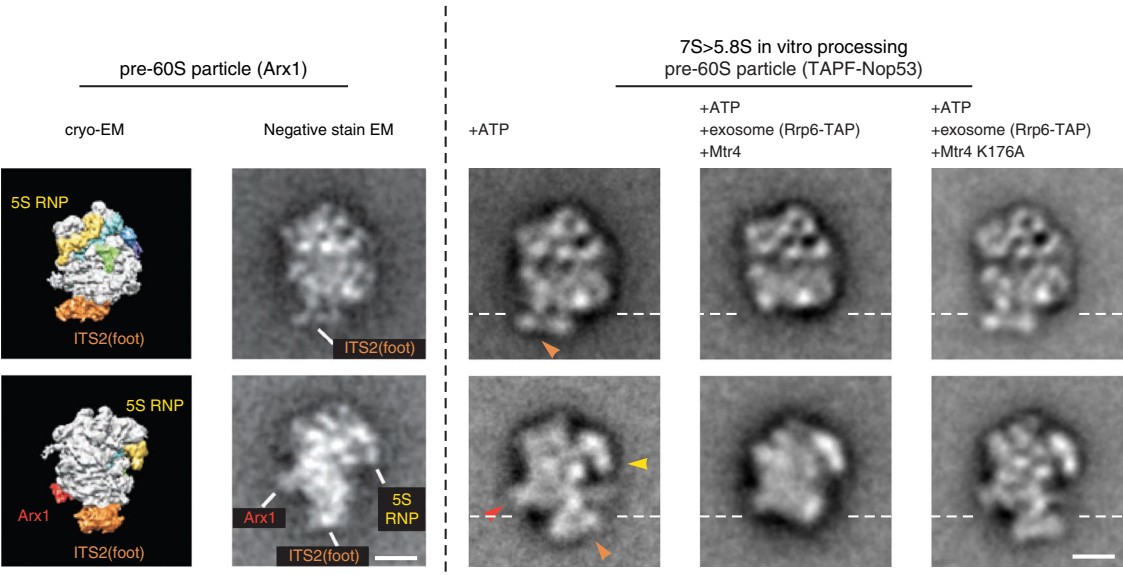

**Fig. 4** Detection of 'foot' structure removal from pre-60S particles. Negative-stain EM of the pre-60S particles (affinity-purified via TAPF-Nop53) after the in vitro processing reaction using endogenous yeast exosome (affinity-purified via Rrp6-FTpA) and recombinant wt Mtr4 or mutant Mtr4 (K176A), all in the presence of ATP. For control purposes, the pre-60S particle was also incubated with only ATP (mock). Shown to the left of the dashed line is the cryo-EM and negative-stain EM structure of the published Arx1-TAP particle (reproduced from ref. [25]) in two orientations, rotated by 90° around the z-axis[25, 26], which reveal structural hallmarks of the 7S pre-rRNA containing pre-60S particle: 'foot' (orange; ITS2 part of 7S pre-rRNA plus ITS2-associated factors), 5S RNP (yellow), Arx1 (red). Shown on the right are the two typical negative-stain EM classes of the pre-60S particles after the in vitro processing reaction, which correspond to the shown classes of the Arx1 particle. The pre-60S particles shown were derived from fraction 8 of the glycerol gradient (Fig. 3a). The orange, yellow and red arrowheads point to ITS2 (foot), 5S RNP and Arx1 structures, respectively. The white dashed horizontal lines mark where the 'foot' structure was removed from the core of the pre-60S particle upon in vitro 7S→5.8S pre-rRNA processing. The scale bars represent 10 nm. See Supplementary Fig. 4 for more class averages

structure, although Arx1 and the 5S RNP remained unaffected (Fig. 4). In contrast, the 'foot' structure of the pre-60S particle remained clearly visible in all of the particles, if the Mtr4 (K176A) mutant was used. Thus, in vitro processing of the 7S→5.8S pre-rRNA induces concomitant removal of the 'foot' structure, which apparently can occur independently of 5S RNP rotation (see the Discussion).

**Reconstitution of all ITS2 removal steps**. As an ultimate goal, we wanted to reconstitute the entire cycle of ITS2 processing, solely based on purified and reconstituted processing factors. For this purpose, we combined the purified Las1–Grc3–Rat1–Rai1 complex, required for ITS2 $C_2$ cleavage and subsequent 26S→25S' pre-rRNA processing, with the fully reconstituted exosome that performs 7S→5.8S pre-rRNA processing (Fig. 5a). If all these components and ATP were added to a pre-60S particle that contained non-cleaved ITS2 substrate RNA (Rsa4 purified from Las1 depletion background carrying 27SB pre-rRNA), the entire series of ITS2 processing steps could be reconstituted (Fig. 5b, c, lane 8). Importantly, upon omitting the Las1 complex, 27SB pre-rRNA was not cleaved and hence no further processing occurred (Fig. 5b, c, lanes 5 and 6). By leaving out specifically one or the other processing complex, we could verify the different and sequential steps, as anticipated from our earlier results. For instance, by adding only the Las1 complex, formation of 26S and 7S intermediates as a result of $C_2$ cleavage was observed (Fig. 5b, c, lane 3), but addition of ATP allowed further 26S→25S' pre-rRNA processing through phosphorylation and thereby activation of the 26S pre-rRNA by Grc3 (Fig. 5b, c, lane 4). On the contrary, if only exosome and cofactors were present and not the Las1 complex, 27SB pre-rRNA remained stable (Fig. 5b, c, lanes 5 and 6). If both Las1 complex and exosome were combined in the

absence of ATP, $C_2$ cleavage occurred but 26S was no longer processed (Fig. 5b, c, lane 7) and also no further 7S pre-rRNA processing occurred. This is consistent with the dependence on ATP of the helicase activity of Mtr4, and that activity being a prerequisite for 7S→5.8S pre-rRNA processing (Fig. 5b, c, lane 7). Hence, addition of ATP restored both 26S→25S' pre-rRNA and 7S→5.8S pre-rRNA processing (Fig. 5b, c, lane 8). If the exosome and co-factors with mutated ribonuclease activities were tested, we observed the same pattern of anticipated intermediates as observed in our partial 7S→5.8S pre-rRNA processing assays (compare Fig. 2b, lanes 9–13 with Fig. 5c, lanes 9–13). Our combined assay further confirmed that Mtr4 catalytic activity or using the double Rrp44 mutant blocked 7S→5.8S pre-rRNA processing, while the Rrp44 single mutants exhibited only partial processing. Last but not least, mutation of Rrp6 caused formation of the 5.8S + 30 intermediate, repeating the hand over from Rrp44 to Rrp6. Taking all these findings together, the complete pathway of ITS2 processing could be reconstituted with purified factors, demonstrating removal of ITS2 from 27SB pre-rRNA in successive and interdependent in vitro processing reactions.

**In vitro 26S→25S' trimming occurs prior to 7S processing**. Using the complete ITS2 processing assay, we could finally test whether 7S pre-rRNA processing is dependent on 26S→25S' pre-rRNA trimming. As the Grc3 kinase mutant (K176A) cannot phosphorylate the 5' end of 26S pre-rRNA, known to be stimulatory for 26S→25S' pre-rRNA removal by the Rat1 5'→3' exonuclease[12], we could test such a mutual dependence by combining wt exosome (enabling 7S pre-rRNA processing) and purified Las1 complex-carrying mutant Grc3 kinase (mediating $C_2$ cleavage without further 26S trimming). Unexpectedly, 7S pre-rRNA was no longer processed by the active exosome in such an

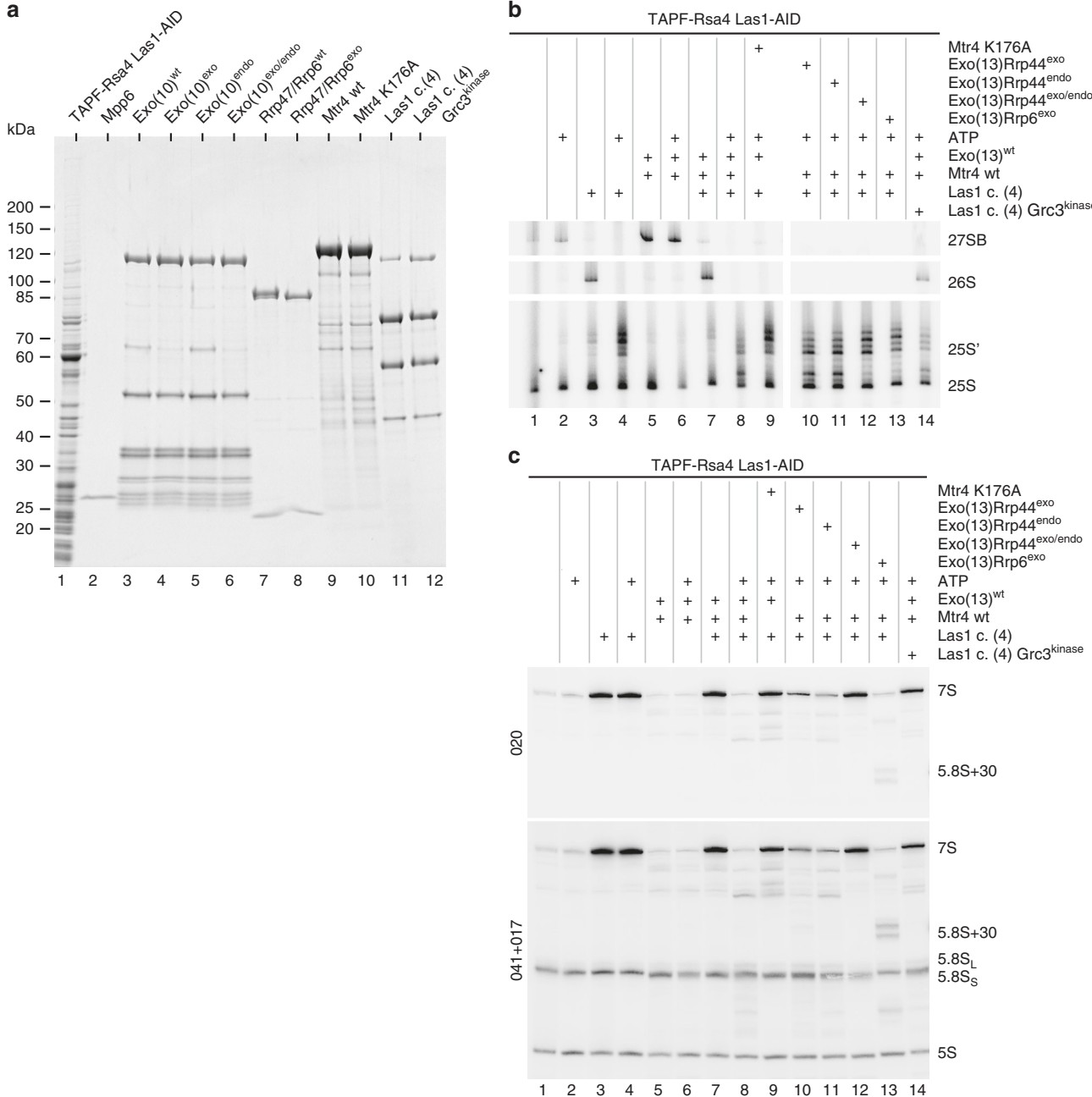

**Fig. 5** Complete cycle of in vitro ITS2 processing by the combined activities of the Las1 complex and nuclear exosome. **a** Analysis of the protein samples used in the complete in vitro ITS2-processing assay. Substrate 27SB pre-rRNA was introduced by addition of pre-60S particles affinity-purified via TAPF-Rsa4 from a *LAS1-AID* degron strain. The yeast exosome subunits and Mtr4 were expressed in *E. coli* and affinity-purified from bacterial cell lysates. Exo(10) comprises the exosome core plus Rrp44. The Las1 complex (Rat1, Grc3, Las1, Rai1, from top to bottom) was overexpressed in yeast and TAP-purified via the split-tag approach (Las1-TpA/Grc3-Flag), yielding a wt and a mutant Las1 complex with a defective Grc3 kinase. The indicated eluates were analysed by SDS-PAGE and Coomassie staining. **b** Primer extension for detection of 27SB pre-rRNA, 26S pre-rRNA, 25S' pre-rRNA and 25S rRNA. **c** Northern blotting for detection of 7S pre-rRNA, 5.8S rRNA and 5S rRNA after the in vitro processing reaction. Processing of 27SB substrate pre-rRNA present in pre-60S particles (affinity-purified via TAPF-Rsa4 from *LAS1-AID degron* strain) using the recombinant exosome and its cofactor Mtr4, and affinity-purified Las1 complex, either wt or harbouring the Grc3 kinase mutant. The in vitro reactions contained the purified proteins and ATP as indicated. 7S pre-rRNA was detected with two different probes. 5S rRNA served as a loading control

in vitro assay, where 26S trimming was blocked (Fig. 5b, c, lane 14). These data indicate that 26S→25S' pre-rRNA processing has to occur first before the nuclear exosome can attack the 7S pre-rRNA for further trimming (see the Discussion).

## Discussion

In this study, we developed an in vitro assay for 7S→5.8S pre-rRNA processing based on purified factors, which gave insight into the series of events occurring during exosome-catalysed

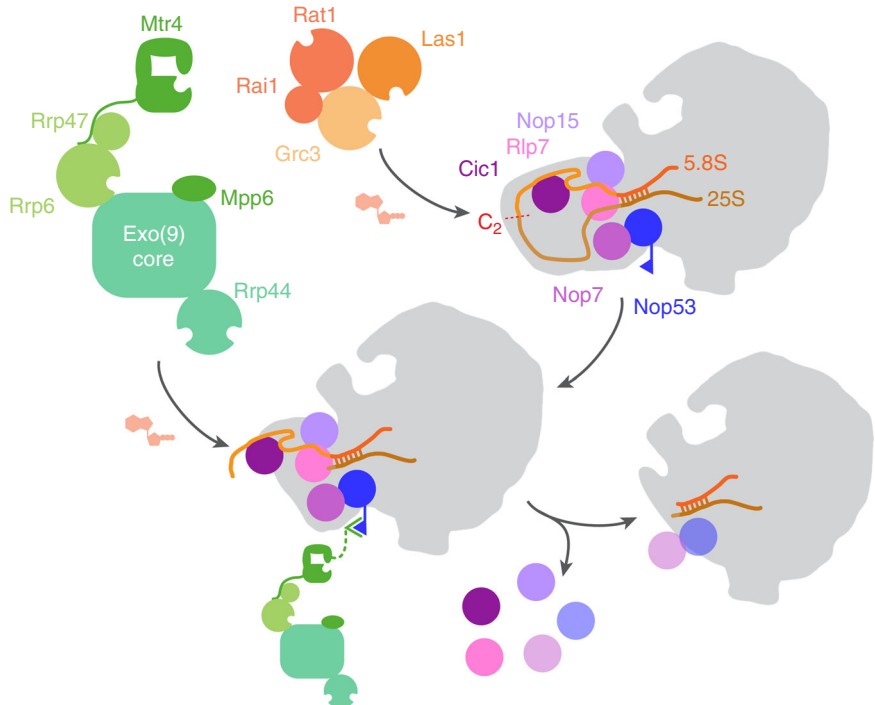

**Fig. 6** Scheme of ITS2 removal from pre-60S particle based on in vitro $C_2$ cleavage and 26S→25S′ pre-rRNA trimming by the Las1 complex, and 7S→5.8S pre-rRNA processing by the nuclear exosome. The Las1 complex initiates ITS2 processing by $C_2$ cleavage and degrades the resulting 26S pre-rRNA to 25S′ pre-rRNA, which facilitates processing of the 7S pre-rRNA by the exosome. The exosome interacts with the pre-60S particle via Mtr4 interaction with Nop53. Mtr4 interacts with a composite surface of Rrp47 and Rrp6 on the exosomal side. Processing of 7S pre-rRNA releases the ITS2-associated factors like Cic1 and Rlp7 efficiently, or like Nop53 and Nop7 only partially

pre-rRNA maturation on pre-ribosomal substrate particles. The ribonuclease activities of the nuclear exosome (Rrp44, Rrp6) were both necessary and sufficient for this reaction to take place, as was the cofactor activity provided by the ATP-dependent RNA helicase Mtr4 that is physically linked to both the exosome and the adaptor protein Nop53 on the pre-60S particle.

In our in vitro assays, we observed that the majority of 7S pre-rRNA is processed to 5.8S rRNA, but a minor signal pool is also detected above the 5.8S signal. In vivo, the exosome is known to trim the 7S pre-rRNA to a 6S intermediate (i.e., 5.8S rRNA carrying eight extra nucleotides), which is exported to the cytoplasm as part of a pre-60S particle, where final processing to 5.8S rRNA is achieved by several cytoplasmic exonucleases (see the Introduction). 6S pre-rRNA would be detectable in the upper northern blot panels. However, if the 6S pre-rRNA is shortened by a few nucleotides, it can only be detected in the lower northern blot panels, in which also slightly elongated species of 5.8S rRNA are detected as well (see Supplementary Fig. 1 for the location of the probes). Therefore, the signal above the 5.8S rRNA species does not correspond to 6S pre-rRNA and we conclude that in our in vitro assay the exosome is able to process 6S pre-rRNA, the majority into 5.8S rRNA and the rest only partially. In vivo this can be avoided by protection of 6S pre-rRNA upon binding of additional ribosome assembly factors and export factors at this site. Notably, one candidate protein is the nuclear export factor Mex67, which was shown to bind in vivo to the 3′ end of the 5.8S rRNA and also to the last six nucleotides within the 6S pre-rRNA[36]. Moreover, Mex67–Mtr2 was shown to bind in vitro to the 3′ end of 5.8S rRNA, if bound to the export-competent Yvh1 particle[37].

In contrast to in vivo data[33, 38], we observed that the endonuclease activity of Rrp44 contributes significantly to in vitro 7S pre-rRNA processing. However, it is possible that in an in vitro situation, in which the major exoribonuclease site of Rrp44 has been mutated, the endoribonuclease site takes over and functions as an exoribonuclease to degrade RNAs threaded via the central channel of the exosome barrel[39–41]. In addition, compared with the single mutants, the effect of the double mutation has been observed to be synergistic in vivo[33, 38, 42].

In agreement with their location in the 'foot' structure, we found that 7S pre-rRNA processing induces dissociation of ITS2-associated factors that are in direct contact with ITS2 rRNA, most prominently Cic1 and Rlp7, and to a lower extent also Nop7. Nop7 is, however, not very stably associated with the pre-60S particle, even in the presence of the catalytically inactive exosome. Consistent with this finding, partial dissociation of Nop7 upon sucrose centrifugation of pre-60S particles was observed in an earlier study[43].

Another observation in our in vitro assay was that 7S→5.8S pre-rRNA processing and 'foot' structure removal are not coupled with 5S RNP rotation. Indeed, we have speculated that the 5S RNP rotation is coupled with Rix1 factor recruitment and Rpf2–Rrs1 dissociation[31, 44]. To what extent removal of the 'foot' structure and 5S RNP rotation are linked in vivo thus remains unclear[3, 45]. In a recent cryo-EM structure of different Nog2 particles, one pool still carried the 'foot' structure, but the 5S RNP was already rotated[30]. Thus, it appears that both events can happen independently of each other. Nevertheless, there is indication of cross-talk between ITS2 processing and 5S RNP rotation in vivo. For example, the hierarchically assembled B-factors are necessary for ITS2 processing, but are located far away from the 'foot' structure, some of them near the 5S RNP[30, 46]. Moreover, mutants were identified that impaired both ITS2 processing and 5S RNP rotation, such as an N-terminal Rpl8 truncation or a Rix1 mutant with impaired Rea1 recruitment[31, 47]. Thus, 7S pre-rRNA processing and 5S RNP rotation occur within a relatively narrow time window during 60S biogenesis, but might not be absolutely coupled and hence can occur independently.

At last, we succeeded in facilitating the entire ITS2 processing reaction in the test tube by combining the two different enzymatic activities required for coupling $C_2$ cleavage with 26S→25S' pre-rRNA processing, and subsequent 7S→5.8S pre-rRNA processing (Fig. 6). From our previous experiments, we knew that pre-60S particles produced by our in vitro $C_2$ cleavage assay carried a cyclic phosphate at the 3′ end of 7S pre-rRNA. We observed that under our in vitro 7S pre-rRNA processing conditions, this did not pose problems to the recombinant exosome. This is in agreement with recent structural data on the exosome, indicating that a cyclic phosphate could be theoretically accommodated in the catalytic centre of Rrp44[15].

Unexpected, however, was another finding that upon blockage of 26S→25S' pre-rRNA trimming in the presence of a Grc3 kinase mutant, subsequent 7S pre-rRNA processing could not occur, although the exosome was fully active. Notably, Rai1 knockout and Rat1 depletion have both been shown to increase 7S pre-rRNA levels in vivo[48], as well as the expression of Grc3 mutants[11]. Las1 has also been implicated in the coordination of processing of both pre-rRNA ends[8]. According to our results, interfering or slowing down of 26S pre-rRNA processing will also affect 7S pre-rRNA processing. The reason for this is unclear, but the 26S pre-rRNA might also reduce the accessibility of the 3′ end of the 7S pre-rRNA, so that the exosome cannot effectively attack it via Mtr4 and Rrp44. Alternatively, some of the ITS2-associated factors might mask the 7S pre-rRNA end if 26S pre-rRNA is not trimmed.

Taken together, our developed in vitro assays enabled us to dissect ITS2 removal in the test tube and gave mechanistic insight into the series of coupled reactions, regarding the role of the different RNA nuclease activities in 7S pre-rRNA processing and the timing of 26S and 7S pre-rRNA processing. Thus, ITS2 processing in vitro was found to be complex and interdependent, requiring several activities of different RNA processing machines. However, overall it is less complicated than previously thought, considering the wealth of in vivo data that have been published on this complex biological problem[46, 49].

## Methods

**Yeast strains and plasmids.** *Saccharomyces cerevisiae* strains used in this study are listed in Supplementary Table 1. The plasmids used in this study are listed in Supplementary Table 2.

**Purification from yeast.** Pre-ribosomal particles and endogenous Rrp6 were genetically encoded with a combination of protein A, a TEV cleavage site and a Flag tag. Cells Cells were lysed using a bead beater and the lysate was cleared by centrifugation in buffer containing 100 mM NaCl, 50 mM Tris (pH 7.5), 1.5 mM MgCl₂, 5% (v/v) glycerol, 1 mM DTT, protease inhibitor (PI) mix and 0.1% (v/v) NP-40. The cleared lysate was incubated with IgG beads for 1.5 h and the beads were washed and further purified in the same buffer without DTT and PI and with only 0.01% NP-40. After TEV elution at 16 °C for 1.5 h, the eluate was incubated with Flag beads for 45 min at 4 °C. The beads were washed and eluted with Flag peptide for 45 min at 4 °C. The eluates were immediately used in the processing assays. Purification of the Las1 complex was described[12]. The complex was stored at −80 °C after purification.

**Purification from E. coli.** *S. cerevisiae* Mtr4 was expressed in *E. coli* BL21 cells. Cells were grown to OD = 0.5, moved to a 23 °C environment, induced with 0.5 mM IPTG for 2 h and harvested. Cells were lysed using a microfluidizer in buffer containing 100 mM NaCl, 50 mM Tris (pH 8), 1.5 mM MgCl₂, 5% glycerol, 2 mM DTT, PI and 0.1% NP-40. The lysate was cleared by centrifugation and incubated with Ni–NTA beads for 1 h at 4 °C in the presence of 10 mM imidazole. Beads were washed in the same buffer without PI and with only 0.01% NP-40. This buffer was also used if the eluate was applied to gel filtration (Superdex 200). Fractions containing Mtr4 were identified by SDS-PAGE and the protein was stored at −80 °C. Recombinant exosome components were purified as previously described[13]. Mpp6 was purified as previously described[20]. Proteins were stored at −80 °C.

**7S pre-rRNA processing assays.** Reaction mixtures were prepared as indicated, and if necessary the volume was adjusted. In the case of the pre-60S particles and exosome purified by Rrp6-FTpA, Flag eluates corresponding to one litre of culture were used for each condition. Mtr4 and ATP were used at concentrations of 10.8 pmol and 0.5 mM, respectively. The assay buffer contained 100 mM NaCl, 50 mM Tris (pH 7.5), 1.5 mM MgCl₂ and 5% glycerol.

In assays with the recombinant exosome, 3.6 pmol of each component was used, except for Mtr4 (10.8 pmol) and Mpp6 (4.3 pmol). ATP was used at a concentration of 0.5 mM. The assay buffer contained 100 mM NaCl, 50 mM Tris (pH 7.5), 3 mM MgCl₂, 5% glycerol and 0.01% NP-40. After 45 min of incubation at 30 °C, the assay was stopped by phenol/chloroform/isoamyl alcohol extraction of the RNA, which was again chloroform-extracted and precipitated. RNA was analysed by northern blotting using the 020 probe (ITS2 5′), TGAGAAGGAAATGACGCT; 017 probe (5.8S), GCGTTCTTCATCGATGC; and 041 probe (5S), CTACTCGGTCAGGCTC.

**ITS2 processing assay.** The assay was performed as described in Gasse et al.[12], 2015, with addition of recombinant exosome components as described for the 7S processing assay in the indicated conditions. The assay buffer contained 100 mM NaCl, 50 mM Tris (pH 7.5), 3 mM MgCl₂, 5% glycerol and 0.01% NP-40. After 30 min of incubation at 30 °C, the assay was stopped, the RNA was extracted and detected by northern blotting as described above. For detection of 25S species primer extension was performed with the 007 probe (25S) CTCCGCTTATTGA-TATGC. Uncropped scans of blots are depicted in Supplementary Fig. 5.

**Glycerol gradient centrifugation.** The assays were performed as described but scaled up with eightfold higher amounts of all components. After incubation, loading samples for RNA and protein analysis were taken and subjected to linear glycerol gradient centrifugation in 100 mM NaCl, 50 mM Tris (pH 7.5), 3 mM MgCl₂, 10–40% glycerol and 0.01% NP-40. Centrifugation was then performed for 14 h at 68027 g. Fractions were collected and either used for negative-stain EM (7.5%) and trichloroacetic acid (TCA) precipitation (92.5%) followed by PAGE and western blotting analysis, or the complete fraction was TCA-precipitated, with half analysed by PAGE and half used for mass spectrometry. In case of the western blots the following antibodies were used followed by their dilutions and source in brackets: α-RpL3 (1:5000, J. R. Warner), α-Cic1 (1:5000, E. Tosta), α-Nop7 (1:60000, B. Stillman), α-rabbit (1:2000, Biorad 170–6515). Uncropped scans of blots are depicted in Supplementary Fig. 5.

**Negative-stain electron microscopy.** Negative staining, data collection and processing were performed as described previously[12]. In brief, 5 μL of the sample was placed on a freshly glow-discharged, carbon-coated grid, allowed to absorb onto the carbon, washed three times with water, stained with uranyl acetate (2% w/v) and dried. Micrographs were recorded using an electron microscope (Tecnai F20, FEI) operating at 200 kV with a bottom-mounted 4 K, high-sensitivity charge-coupled device camera (Eagle, FEI) at a nominal magnification of 29000 (calibrated pixel size of 3.81 Å). For averaging, 3908 particles for the TAPF-Nop53 + ATP (Mock) sample, 4414 particles for the TAPF-Nop53 + Rrp6-FTpA + Mtr4 + ATP sample, and 3870 particles for the TAPF-Nop53 + Rrp6-FTpA + Mtr4 K176A + ATP sample were selected using the auto-boxing feature of EMAN2[50]. Image processing was carried out using the IMAGIC-4D package[51]. Particles were band-pass filtered and normalised in their grey value distribution, and mass-centred. Two-dimensional alignment, classification and iterative refinement of class averages were performed as described previously[52].

**Mass spectrometry and data processing.** Label-free quantification (LFQ) was performed using 1D *n*LC–MS/MS at the FingerPrints Proteomics Facility at the University of Dundee. Data were analysed using MaxQuant software with the SGD database[53]. Output LFQ values of ribosome biogenesis factors were normalised to Nog1 and plotted against each other (see Fig. 3b and Supplementary Table 3). A list of LFQ values and proteins identified in two experiments can be found in Supplementary Table 3.

**Data availability.** All relevant data supporting the findings of this study can be found in the results or the supplementary information section, or are available from the corresponding author upon request. All experiments were performed at least twice with similar outcome. The data of these biological replications are available from the authors upon reasonable request.

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

## Acknowledgements

We thank Jochen Bassler for support in the analysis of the LFQ data. We thank J. R. Warner (anti-Rlp3), B. Stillman (anti-Nop7) and E. Tosta (anti-Cic1) for the kind gift of antibodies. E.H. is a recipient of grants from the Deutsche Forschungsgemeinschaft (DFG; HU363/10-5, HU363/12-1).

## Author contributions

L.F. and E.H. designed the study in collaboration with E.C. L.F. performed all in vitro assays, glycerol gradient centrifugations, and protein and RNA analyses. S.F. and J.M.S.

provided the recombinant exosome. D.F. performed negative-stain EM. M.T. provided strains and Mtr4-expression plasmids. L.F. and E.H. wrote the manuscript. All authors discussed the results and commented on the manuscript.

## Additional information

**Competing interests:** The authors declare no competing financial interests.

