## [Peer Review File · Nature Communications]

Reviewers' comments:

Reviewer #1 (Remarks to the Author):

Here Fromm et al. have extended their previous analysis of the Las1 complex (Gasse et al. 2015) to study ITS2 processing. The authors show that purified factors can be used to reconstitute processing of a 27S pre-rRNA intermediate to 25S and 5.8S rRNAs. Mutant forms of different enzymes have further been used to dissect their specific functions in this pathway. The presented findings are clearly illustrated and of general interest to the scientific community. This manuscript should therefore be published in Nature Communications once the following points have been addressed.

Major points

1. In the summary the authors claim that the ITS2 in pre-ribosomal RNA is “reminiscent of an intron that needs to be removed to make functional ribosomes”.

ITS2, similar to other spacers that are present in pre-ribosomal RNA (for example ITS1), is simply an internal RNA segment that is enzymatically removed during ribosome assembly. The underlying mechanisms that are employed for ITS2 processing and splicing are conceptually very different.

Since there is no scientific basis for the comparison of splicing and ITS2 processing, the authors should remove these references (pages 2 and 3) in the manuscript.

2. The authors claim that “approximately 90% of particles have lost their foot structure, although the knob and nose remained unaffected.”

First, from the few selected 2D class averages it is unclear how homogenous the sample was. Since several thousand particles were used for 2D classification, the authors should provide supplementary figures that show all 2D class averages that were obtained from these experiments.

Second, the authors use terms such as “foot”, “knob” and “nose”, which describe features such as ITS2, Arx1 and the 5S RNP. Since the molecular basis for these features is known, ITS2, Arx1 and 5S RNP should be used for clarity instead.

3. It is currently unclear which probes were used for the Northern blots that indicate the 7S pre-rRNA in two instances (Fig. 1b, 2b, 5c). The identities and positions of these probes should be indicated in Supplementary Figure 1 for clarity.

4. On page 13 the authors state:

“Unexpectedly, we found that 7S pre-rRNA processing induces dissociation of ‘foot’ binding proteins that are in direct contact with ITS2 rRNA, most prominently Nsa3 and Rlp7, and to a lower extent also Nop7.”

The structure of the Nog2 particle (Wu et al.) has clearly shown that ITS2 and its associated factors Cic1, Rlp7 and Nop7 are in close proximity to 5.8S rRNA and hence its precursor, the 7S pre-rRNA.

It is therefore absolutely expected that processing of a 7S pre-rRNA would affect the neighboring regions, such as ITS2. This sentence should be changed accordingly.

5. In the discussion (page 14) the authors claim that the in vitro assays gave mechanistic insight “on a even higher level, the timing of ITS2 rRNA processing and 5S RNP rotation.”

Since the authors state in the discussion: “To what extent removal of the foot structure and 5S RNP rotation are linked in vivo thus remains unclear”, claiming that the current manuscript provides “insights into the timing of ITS2 processing and 5S RNP rotation” is inconsistent and should be removed.

Minor points

1. Many proteins are involved in ribosome assembly and unfortunately different names exist for multiple proteins. This should be kept to an absolute minimum. It would be helpful if Cic1 and not Nsa3 could be used throughout this manuscript, since Cic1 is the standard name of this protein, which was also used by Wu et al. who first described its structure.

2. Similarly, as stated above, the authors should minimize the use of other terms, such as “foot”, which simply describes ITS2 and associated factors. This is particularly relevant on page 8, where this terminology (“foot factors”) is used in a title.

3. The molecular weight markers are only partly labeled and should be indicated in full.

Reviewer #2 (Remarks to the Author):

Ribosomal RNA mature ends are produced by extensive processing. Two of the three rRNAs of the large ribosomal subunit (60S), the 5.8S and 25S rRNAs, are separated by an internal transcribed spacer (ITS2) that requires precise excision. The authors have addressed a specific step of pre-rRNA processing (ITS2 removal) in budding yeast, using a combination of in vitro reconstitution assays and low-resolution cryoEM.

They successfully recapitulated formation of the 5.8S 3' end and 25S rRNA 5' end, and concluded that ITS2 removal coincides with the disappearance of a so-called 'foot' structure, normally visible by cryoEM on purified pre-60S ribosomes.

Although this work presents elegant biochemistry, in my opinion, there is insufficient novelty to grant publication in Nature Communications: the processing enzymes, cleavage steps, sequence of cleavages, and 'ITS2-containing foot' were all known. The work addresses a very specific step of pre-rRNA processing with no general principles that could be extrapolated to other systems (outside what was already known, i.e. exosomal subunit handover, and coordination of processing at both ends of ITS2 – see below).

Comments

1) A model for coordinated pre-rRNA processing at both ends of ITS2 involving a Las1-containing complex has been proposed in both yeast and human cells (Schillewaert et al 2012 PMID 22083961). This model should be discussed in the introduction of this paper. The authors know this model very well and should refer to it fully. I can only assume this is a fortuitous omission. Generally, former work on Las1 is also very poorly referenced (apart from the reference from the authors). Prof Catherine Dénicourt (Texas) published several very useful contributions, this submission would also benefit from listing them.

2) Fig 2, panel b, some lanes have less 7S than others, indicating as discussed by the authors, partial processing by the partially inactivated Rrp44 (lanes 10 and 11). This calls for a systematic quantification of all major bands shown. This will also help understanding if the ratio of 5.8 short versus long is affected or not (by the way, 5.8S should be labelled for its two forms).

Reviewer #3 (Remarks to the Author):

Reviewer's comments, Fromm et al., NCOMMS-17-17168-T

In the submitted manuscript, Fromm and colleagues reconstitute the yeast ITS2 processing, using a recombinantly expressed exosome complex as well as Las1 complex, in addition to required additional factors such as Mtr4 and Grc3 on substrate precursor ribosomal RNA - either 27SB or 7S – isolated as part of a Nop53 complex purified from yeast extract. The authors showed that i) both Rrp44 and Rrp6 are required for 7S-5.8S processing and that their function is consecutive, Rrp44 first, followed by Rrp6; ii) that Mtr4 is required for 7S-5.8S processing. In addition, the authors show that 7S-5.8S processing causes the release of most of the 'A3 cluster' proteins, and the removal of the pre-60S foot structure. Moreover, the authors show that 26S-25S' processing has to occur for 7S-5.8S processing to commence, possibly due to blocking the entry site to the 3'end of 7S for the exosome.

This manuscript presents very nice work of high technical quality. The recombinant complexes are clean and all experiments are well controlled. The data presented here brings together nicely previously published in vivo phenotypical processing, proteomic, and structural data in a number of elegant biochemical experiments, demonstrating the previously suggested concerted action of RNA processing machineries and complexes, and consequently illuminating part of a complex

cellular process, ribosome biogenesis. I would recommend the manuscript for publication in Nature Communications.

Minor points:

- Page 12, 2nd paragraph; please make clear that the ‘slightly elongated species of 5.8S’ that are detected by the probe mentioned here, are not detecting 6S due to it being processed in the in vitro assay and the absence to factors that may block its processing in vivo (Mex67-Mtr2 or else). The explanation was somewhat, confusing.
- Page 10, throughout the page the authors discuss both data from Fig.5 panels and c, however, only panel b is referred to, which makes it confusing. Please change to “Fig. 5b and c” throughout.
- Page 12, 2nd paragraph, line 3: “In vivo, the exosome...” please add comma.
- Page 13, line 6: “Nop7 is, however, not very...” please add commas.
- page 29: change “7S pre-rrna” to “7S pre-rRNA”

Point-to-point reply to the Reviewers' comments:

Reviewer #1 (Remarks to the Author):

Here Fromm et al. have extended their previous analysis of the Las1 complex (Gasse et al. 2015) to study ITS2 processing. The authors show that purified factors can be used to reconstitute processing of a 27S pre-rRNA intermediate to 25S and 5.8S rRNAs. Mutant forms of different enzymes have further been used to dissect their specific functions in this pathway.

The presented findings are clearly illustrated and of general interest to the scientific community. This manuscript should therefore be published in Nature Communications once the following points have been addressed.

Major points

1. In the summary the authors claim that the ITS2 in pre-ribosomal RNA is "reminiscent of an intron that needs to be removed to make functional ribosomes".

ITS2, similar to other spacers that are present in pre-ribosomal RNA (for example ITS1), is simply an internal RNA segment that is enzymatically removed during ribosome assembly. The underlying mechanisms that are employed for ITS2 processing and splicing are conceptually very different.

Since there is no scientific basis for the comparison of splicing and ITS2 processing, the authors should remove these references (pages 2 and 3) in the manuscript.

We have removed this comparison between ITS2 removal and intron splicing from the abstract

2. The authors claim that "approximately 90% of particles have lost their foot structure, although the knob and nose remained unaffected."

First, from the few selected 2D class averages it is unclear how homogenous the sample was. Since several thousand particles were used for 2D classification, the authors should provide supplementary figures that show all 2D class averages that were obtained from these experiments.

We have now included all 2D averages in Supplementary Figure 4.

Second, the authors use terms such as "foot", "knob" and "nose", which describe features such as ITS2, Arx1 and the 5S RNP. Since the molecular basis for these features is known, ITS2, Arx1 and 5S RNP should be used for clarity instead.

We have replaced "nose" with 5S RNP, "knob" with Arx1 and "foot" with ITS2. However, we sometimes use the term "Foot" in the text, which we initially created to describe a typical hallmark structure of the Arx1 pre-60S particle, visible by negative stain EM. This allows us to keep some simplicity in the text as opposed to describing all the factors contained in the "foot" every time with "part of ITS2 that is part of 7S pre-rRNA and ITS2 associated factors". However, we changed all mentioning of "foot factors" and alike to "ITS2 associated factors", although this requires some flexibility in the usage of "associated", since two of the proteins do not directly associate with/bind to ITS2.

3. It is currently unclear which probes were used for the Northern blots that indicate the 7S pre-rRNA in two instances (Fig. 1b, 2b, 5c). The identities and positions of these probes should be indicated in Supplementary Figure 1 for clarity.

We have changed the labeling of all northern blots to indicate the probes used on the left side of the panel and indicated the location of the probes in Supplementary Figure 1, as suggested.

4. On page 13 the authors state:

"Unexpectedly, we found that 7S pre-rRNA processing induces dissociation of "foot" binding proteins that are in direct contact with ITS2 rRNA, most prominently Nsa3 and Rlp7, and to a lower extent also Nop7."

The structure of the Nog2 particle (Wu et al.) has clearly shown that ITS2 and its associated factors Cic1, Rlp7 and Nop7 are in close proximity to 5.8S rRNA and hence its precursor, the 7S pre-rRNA.

It is therefore absolutely expected that processing of a 7S pre-rRNA would affect the neighboring regions, such as ITS2. This sentence should be changed accordingly.

We have changed the sentence and now say "In agreement..."

5. In the discussion (page 14) the authors claim that the *in vitro* assays gave mechanistic insight "on an even higher level, the timing of ITS2 rRNA processing and 5S RNP rotation."

Since the authors state in the discussion: "To what extent removal of the foot structure and 5S RNP rotation are linked *in vivo* thus remains unclear", claiming that the current manuscript provides "insights into the timing of ITS2 processing and 5S RNP rotation" is inconsistent and should be removed.

This sentence has now been changed. We now say "Taken together, our developed *in vitro* assays enabled us to dissect ITS2 removal in the test tube and gave mechanistic insight into the series of coupled reactions, regarding the role of the different RNA nuclease activities in 7S pre-rRNA processing and the timing of 26S and 7S pre-rRNA processing.

Minor points

1. Many proteins are involved in ribosome assembly and unfortunately different names exist for multiple proteins. This should be kept to an absolute minimum. It would be helpful if Cic1 and not Nsa3 could be used throughout this manuscript, since Cic1 is the standard name of this protein, which was also used by Wu et al. who first described its structure.

Nsa3 was replaced by Cic1 in all instances.

2. Similarly, as stated above, the authors should minimize the use of other terms, such as "foot", which simply describes ITS2 and associated factors. This is particularly relevant on page 8, where this terminology ("foot factors") is used in a title.

We have changed all mentioning of "foot factors" and alike to "ITS2 associated factors", see comment on major point 2.

3. The molecular weight markers are only partly labeled and should be indicated in full.

All markers were fully labeled.

Reviewer #2 (Remarks to the Author):

Ribosomal RNA mature ends are produced by extensive processing. Two of the three rRNAs of the large ribosomal subunit (60S), the 5.8S and 25S rRNAs, are separated by an internal transcribed spacer (ITS2) that requires precise excision. The authors have addressed a specific step of pre-rRNA processing (ITS2 removal) in budding yeast, using a combination of *in vitro* reconstitution assays and low-resolution cryoEM.

They successfully recapitulated formation of the 5.8S 3' end and 25S rRNA 5' end, and concluded that ITS2 removal coincides with the disappearance of a so-called "foot" structure, normally visible by cryoEM on purified pre-60S ribosomes.

Although this work presents elegant biochemistry, in my opinion, there is insufficient novelty to grant publication in Nature Communications: the processing enzymes, cleavage steps, sequence of cleavages, and 'ITS2-containing foot' were all known. The work addresses a very specific step of pre-rRNA processing with no general principles that could be extrapolated to other systems (outside what was already known, i.e. exosomal subunit handover, and coordination of processing at both ends of ITS2 (see below).

We respectfully disagree with this reviewer's statement regarding the novelty of our findings. While it is true that the processing enzymes and cleavage steps are known, these were so far shown *in vivo*. Here we show *in vitro* the sole dependency of the 7S pre-rRNA processing on the activities of Rrp44, Rrp6 and Mtr4. We could not only observe this on rRNA level but also on the structural level by protein dissociation and the disappearance. While it is expected that this somehow must happen, being able to watch this directly is entirely novel. Even processing of 27SB pre-rRNA is possible with these activities in concert with the Las1 complex. This is especially interesting since the processing was proposed to be very complicated depending on other factors and ITS2 conformations. Thus, our *in vitro* assay shows a remarkable simplification of the mechanistic aspects of ITS2 processing.

Comments

1) A model for coordinated pre-rRNA processing at both ends of ITS2 involving a Las1-containing complex has been proposed in both yeast and human cells (Schillewaert et al 2012 PMID 22083961). This model should be discussed in the introduction of this paper. The authors know this model very well and should refer to it fully. I can only assume this is a fortuitous omission. Generally, former work on Las1 is also very poorly referenced (apart from the reference from the authors). Prof Catherine Dénicourt (Texas) published several very useful contributions, this submission would also benefit from listing them.

We thank this reviewer for reminding us about these important publications. Indeed in an earlier version, we referred to the Schillewaert et al. paper, but somehow this citation got lost during the final submission. We now describe these findings in the revised discussion, to correlate our data on the timing of

ITS2 processing with what is known in literature. Moreover, we have added some new references in the Introduction, which deal with the previously identified interaction partners of Las1, including the work from Prof Catherine Dénicourt.

2) Fig 2, panel b, some lanes have less 7S than others, indicating as discussed by the authors, partial processing by the partially inactivated Rrp44 (lanes 10 and 11). This calls for a systematic quantification of all major bands shown. This will also help understanding if the ratio of 5.8 short versus long is affected or not (by the way, 5.8S should be labelled for its two forms).

We have included a quantification of 7S pre-rRNA levels in Supplementary Figure 2a. Quantification of the 5.8S rRNA short versus long form was not possible, since in case of 7S pre-rRNA processing the signal cannot be unambiguously attributed to one of the forms, as discussed in the second paragraph of the discussion. All northern blots are now labeled for the two forms of 5.8S rRNA.

Reviewer #3 (Remarks to the Author):

Reviewer's comments, Fromm et al., NCOMMS-17-17168-T

In the submitted manuscript, Fromm and colleagues reconstitute the yeast ITS2 processing, using a recombinantly expressed exosome complex as well as Las1 complex, in addition to required additional factors such as Mtr4 and Grc3 on substrate precursor ribosomal RNA - either 27SB or 7S isolated as part of a Nop53 complex purified from yeast extract. The authors showed that i) both Rrp44 and Rrp6 are required for 7S-5.8S processing and that their function is consecutive, Rrp44 first, followed by Rrp6; ii) that Mtr4 is required for 7S-5.8S processing. In addition, the authors show that 7S-5.8S processing causes the release of most of the 'A3 cluster' proteins, and the removal of the pre-60S foot structure. Moreover, the authors show that 26S- 25S processing has to occur for 7S-5.8S processing to commence, possibly due to blocking the entry site to the 3'end of 7S for the exosome.

This manuscript presents very nice work of high technical quality. The recombinant complexes are clean and all experiments are well controlled. The data presented here brings together nicely previously published in vivo phenotypical processing, proteomic, and structural data in a number of elegant biochemical experiments, demonstrating the previously suggested concerted action of RNA processing machineries and complexes, and consequently illuminating part of a complex cellular process, ribosome biogenesis. I would recommend the manuscript for publication in Nature Communications.

Minor points:

- Page 12, 2nd paragraph; please make clear that the "slightly elongated species of 5.8S" that are detected by the probe mentioned here, are not detecting 6S due to it being processed in the in vitro assay and the absence to factors that may block its processing in vivo (Mex67-Mtr2 or else). The explanation was somewhat, confusing.

We have now clearly indicated that the observed signal is not 6S pre-rRNA and changed the text accordingly. The location of the probes is now indicated in the Supplementary Figure1 and the probes used are indicated next to the panel.

- Page 10, throughout the page the authors discuss both data from Fig.5 panels and c, however, only panel b is referred to, which makes it confusing. Please change to "Fig. 5b and c" throughout.

The references to Fig. 5c were now introduced.

- Page 12, 2nd paragraph, line 3: "In vivo, the exosome" please add comma.

Comma was added.

- Page 13, line 6: "Nop7 is, however, not very" please add commas.

Commas were added.

- page 29: change "7S pre-rrna" to "7S pre-rRNA"

"7S pre-rrna" was changed to "7S pre-rRNA".

Reviewers' Comments:

Reviewer #1 (Remarks to the Author):

All my points have been addressed by the authors. The revised manuscript should be published in Nature Communications.

Reviewer #2 (Remarks to the Author):

The authors have adequately addressed my comments, and those of the other referees.

This is a nice now scholarly balanced contribution by the Conti/Hurt laboratories.